# The Views on Terrorism in the Name of Islam Held by Islamic Religion Teachers in Spain

**María Navarro-Granados**, **Vicente Llorent-Bedmar**
**and Verónica C. Cobano-Delgado Palma** *

Departamento de Teoría e Historia de la Educación y Pedagogía Social, Facultad de Ciencias de la Educación, Universidad de Sevilla, C/Pirotecnia s/n, 41005 Sevilla, Spain; mnavarro11@us.es (M.N.-G.); llorent@us.es (V.L.-B.)
* Correspondence: cobano@us.es; Tel.: +34-955-42-05-7

**Abstract:** Violent radicalisation is currently one of the most pressing global problems. Accordingly, the intention of this paper is to discover the views on terrorism in the name of Islam held by Islamic religion teachers in Spain and to analyse the preventive socio-educational measures that they propose, employing a mixed methods approach with a questionnaire and semi-structured interviews. Most of the teachers point to a poor knowledge of Islam among the young, especially as regards to the concept of jihad, as the main reason behind this phenomenon. The second-generation immigrants among the teachers attach great importance to possible identity crises as a factor that makes the young more vulnerable to violent radicalisation. The most frequently mentioned social measure is fostering the integration of Muslims, thus creating a sense of belonging to their country of residence. They recommend preventive educational strategies that, far from focusing on detecting possible violent extremists, place the accent on teaching the fundamentals of the Islamic religion in order to provide young people with the tools that they need to challenge violent ideologies.

**Keywords:** Islamic religious education; Muslim teachers; radicalisation; Spain

---

## 1. Introduction

Terrorists attacks carried out in the name of Islam have increased exponentially over the past few years, becoming one of the most worrying social problems at a global level (Cottee 2019; Tankel 2019). Specifically, following the 9/11 terrorist attacks in New York, there was growing concern about the compatibility between Islam and the West (Hussain and Read 2015), an issue that has only been exacerbated by the so-called 'domestic terrorism', namely, acts of violence committed by individuals born and bred in the country in question, especially since the 2005 terrorist attacks in London (Crone and Harrow 2011). Since then, and with the rise of groups like Daesh that, according to authors like Akhmetova and Jaafar (2020), use Islam to justify their acts of violence, there has been a mistaken tendency to see Islam as a religion that promotes violence (Waghid and Davids 2014).

After the fall of Daesh in Iraq and Syria, the group has by no means lost its appeal. Indeed, the majority of the group's sympathisers are still firmly committed to its ideology (Massé 2020). To this should be added the following two dilemmas: (1) the people who have returned from the Syrian territory previously controlled by Daesh (Govier and Boutland 2020); and (2) the children who, according to Alsaleh (2019), have been ideologically indoctrinated by the group.

In recent years, there has been a growing interest in violent radicalisation, focusing all but exclusively on Muslim youths (Gurlesin et al. 2020).

For different authors and international bodies, like the UNESCO, the role of education in preventing violent radicalism has become vitally important (Gurlesin et al. 2020; Mattsson and Säljö 2018;

UNESCO 2017). However, as Panjwani (2016) contends, many of these strategies are not based on a truly educational approach, but leverage education for state security purposes.

Specifically, for authors like Abdullah and Saeed (2016), Rodríguez (2019) and Sjøen and Mattsson (2020), among others, the teaching of Islamic religion plays a very important role in the educational prevention of violent radicalism. It has been shown that groups like Daesh employ an out-of-context theological narrative that is proving to be very influential, especially among the young (Mansouri and Keskin 2019). In this vein, Islamic religion teachers can be presented as role models for their students (Vince 2019), helping them to acquire the necessary skills and capabilities for recognising terrorist propaganda (Brooks and Ezzani 2017).

Few studies have focused on analysing the views on terrorism in the name of Islam and on some Islamic concepts held by Islamic religion teachers in Spain. Bearing in mind the great variety of possible interpretations of the sacred texts and that it is the teaching staff who decide on what interpretation they should teach their pupils, this is a fundamental issue. Nevertheless, the lack of empirical data in this regard means that policies are based more on suppositions and ideologies than on demonstrable facts deriving from research (Mattsson and Säljö 2018).

It order to remedy this gap, the aim here is to become acquainted with the views on terrorism in the name of Islam and on its prevention in the socio-educational setting held by Islamic religion teachers in Spain.

To this end, the research questions are as follows: What are the views of Islamic religion teachers in Spain on terrorism in the name of Islam? What socio-educational measures do they propose? What is their understanding of the concept of jihad and the way in which the sacred texts should be interpreted?

In order to answer these questions, a mixed methods, non-experimental empirical study was performed. This involved administering a questionnaire and conducting semi-structured interviews with an eye to gathering the opinions of Islamic religion teachers in Spain.

## 2. Theoretical Framework

### 2.1. The Power of Words: Terrorism in the Name of Islam

The terminology used to refer to terrorism in the name of Islam is an aspect that should be clarified (Baele et al. 2017), insofar as it is usually employed uncritically, thus offending the Muslim community (Antúnez and Tellidis 2013). Terms like Salafism, fundamentalism, jihadism and radicalism are indiscriminately used to refer to this phenomenon.

Salafism (*salafiyya*) is a traditionalist current whose principal aim is to return to the origins of Islam and which regards the '*Salaf*' (the companions of the Prophet) as the most virtuous generation (Fernández-Montesinos 2015). As the Salafists believe that present-day Islam has distanced itself from its roots, they want to recuperate the pure Islam practiced by those generations (Saada 2018).

Salafism is often defined as a homogeneous current, ignoring its diversity (Husnul 2017). Armed groups like Daesh and Al Qaeda are related to a branch of Salafism: the 'Salafi-jihadi' (*al-Salafiyya al-jihadiyya*). As Antúnez (2017) observes, however, it is a mistake to consider Salafism as an intrinsically violent branch of Islam, for the majority of its followers do not believe that violence is an adequate way of changing the existing political order.

The term 'fundamentalism' is also erroneously associated with violence in the name of Islam (Koopmans 2014). It first emerged in the United States at the beginning of the 20th century with the Christian Protestant current. It defends a literal interpretation of the sacred texts (Gierycz 2020). Although, for many Muslims, it encompasses those who comply with the basic tenets of their faith (Massod 2006).

However, if there is a term that has been totally misinterpreted, ignoring its historical, political and social context, then that is jihad (Frissen et al. 2018). In the Qur'an, a distinction is drawn between greater jihad (*yihad al-akbar*) and lesser jihad (*yihad al-ashgar*) (Silverman and Sommer 2019). The former refers to the effort that all Muslims should make to overcome internal temptations and passions (Cohen 2013);

while the latter refers to the sole legal and legitimate war in Islam (Antúnez and Tellidis 2013). Its origin and meaning are to be found in a specific historical context in which the opponents of the Prophet (belonging to the Arab tribe *Quraysh*) ejected the Muslims from their homes in Mecca (Coady and O'Keefe 2002). Some Muslim scholars are of the opinion that lesser jihad can only be applied to the first generation of Muslims who experienced this specific context (Afsaruddin 2016).

According to authors like Rane (2019), the Prophet never resorted to offensive jihad and it should be understood as resistance to oppression. Be that as it may, Mohd et al. (2017), among others, assert that lesser jihad can be both defensive, when Muslims have been previously attacked, and offensive.

For the UNESCO (2017), radicalisation is a process in which a person adopts extreme opinions or practices, even going so far as to legitimise the use of violence. Nevertheless, for authors such as Neumann (2013), radicalisation can prompt people to take extremist stances, but does not inevitably imply the use of violence. In effect, history is replete with radical movements that have achieved positive changes peacefully (UNDP 2016), such as, for example, Mahatma Gandhi with his concept of *Satyagraha* or steadfast nonviolent opposition.

## 2.2. Background

Research on the reasons behind violent radicalisation is far from being conclusive. Multiple factors have been confirmed, whose incidence can vary from person to person (Lynch et al. 2015). According to the Radicalisation Awareness Network (RAN 2016), these include, among others, individual (exclusion, anger and frustration, feelings of injustice, etc.), social (real or perceived discrimination, poor education, etc.) and political (complaints against Western foreign policy) reasons. These factors have taken the shape of a series of "indicators of radicalisation" in some antiterrorist strategies (Dzhekova et al. 2017).

The relationship between educational attainment and terrorism is not that clear. Authors like Sageman (2004), Krueger and Malečková (2003) and Cherney and Povey (2013) have detected that people with higher academic qualifications are more likely to become active members of terrorist groups. In contrast, for others, including Zhirkov et al. (2014), in Western Europe young people with a lower educational level are more prone to supporting terrorism. For their part, Kamarulnizam et al. (2012) have not found any relationship between academic qualifications and support for terrorism; while according to others, like Brockhoff et al. (2015), lower educational levels are related to a greater support for terrorism when accompanied by unfavourable political, socioeconomic and demographic conditions.

Of all these preventive strategies, doubtless, the most well-known is Prevent, one of the four elements of CONTEST, the United Kingdom's counter-terrorism strategy, which obliges teaching staff to detect possible signs of radicalisation in their pupils (HM Government). Despite references to the violent radicalism of the extreme right, Muslims are its unquestionable objective (Novelli 2017). Strategies of this type run the risk of presenting the Muslim community in a uniform manner, without taking into account its enormous diversity, and treating Muslim pupils as a potential threat (Revell 2019); even more so, considering that there is no sole way of identifying an individual who may be susceptible to being influenced by the ideology of terrorist groups (Department for Education 2015).

In Spain, a preventive strategy of this type has yet to be implemented at a national level. Nonetheless, in the past few years, plenty of attention has been paid to the role that the Islamic religion module can play in preventing terrorism in the name of Islam. Indeed, the ex-president of the CIE (Islamic Commission of Spain), Riay Tatary, underscored how important it was for the young to have a solid Islamic education in order to avoid violent radicalism (UCIDE 2017).

In 2014, the 'rejection of terrorist violence' (BOE 2014, p. 101210) was included as a transversal topic in the Islamic religion curriculum in the primary education stage, and in 2016, a content block specifically designed to address the prevention of terrorism was introduced in the secondary education stage. Such content includes topics such as, for instance, the social factors that may influence radicalisation and the identification of youth with violent radicalisation (BOE 2016).

### 2.3. The Teaching of Islamic Religion in Spain

In Spain, the religion module taught at schools is denominational (Díez de Velasco 2015). The 1978 Spanish Constitution currently in force marked a break with Catholicism as the official religion of Francoism, advocating instead for religious pluralism (Vilà et al. 2019). Nonetheless, this break has inescapable nuances, since explicit reference is made to Catholicism in the constitution itself (BOE 1978, Art. 16.3).

The teaching of religion at Spanish state schools is a complex and very politicised issue, resulting from the prolonged contention between the different political parties, evidenced by the diverse education laws that have been passed during the current democratic period (Dietz et al. 2011). Nowadays, the education law in force (Ley Orgánica de Mejora de la Calidad Educativa) establishes that the religion module be included in the academic records of pupils and, therefore, has an influence when requesting study grants and choosing a university career, an issue on which the political parties disagree (Debón et al. 2019).

Although it is currently obligatory to offer the Catholic religion module at all state primary and secondary schools, it is optional for pupils (LOMCE 2013). The teaching of minority religions is governed by co-operation agreements signed with the state. Specifically, the 1980 Fundamental Law on Religious Liberty (Ley de Libertad Religiosa) established that the state could sign these agreements with those religious communities firmly established in the country (BOE 1980, Art. 7), namely, Catholicism, Evangelism, Judaism and Islam. The co-operation agreement with the Islamic Commission of Spain (hereinafter CIE), the body officially representing Muslims in the country, was signed in 1992. This agreement regulates aspects such as the building of mosques and Islamic cemeteries and the teaching of Islamic religion, among others.

In Spain, Islamic religion teachers are proposed by the CIE, as with module content and textbooks, before being approved by the Ministry of Education (BOE 1992). The requirements with which teachers should comply in order to be engaged as such are established in Royal Degree (Real Decreto 2020, Art. 3) 696/2007, among other pieces of legislation:

1. To hold a degree equivalent to that required of official non-university teachers in the respective educational stage. To teach in the pre- and primary school stages, they should hold a bachelor's degree in pre-/primary school teacher training, and in the secondary education stage, a bachelor's degree and a master's degree in secondary school teacher training.
2. To have been proposed by the CIE and to have obtained the declaration on eligibility or certificate required by this body.
3. To be a Spanish national or a foreigner with a legal Spanish residence permit.

Along with these requirements, however, the CIE can propose those that it deems fit.

Minority religions have a very scant presence at Spanish state schools. In this connection, authors like Garreta-Bochaca et al. (2019) question the capacity of the Spanish denominational model to meet the challenges of a plural society from a religious perspective. At present, there are only 76 Islamic religion teachers for the 312,498 Muslim pupils enrolled at Spanish state schools (UCIDE 2019). In order to teach this module, it is up to the education administration to inform the CIE of the need to engage teachers when at least 10 pupils have requested the module.

## 3. Materials and Methods

### 3.1. Study Design and Population

With the aim of gaining a deeper understanding of the object of study, a mixed methods approach (Molina-Azorín 2009), including quantitative and qualitative data collection techniques, was followed. Both methods were employed in two consecutive stages (Creswell 2013). In the first quantitative stage, the teachers completed a specially tailored questionnaire and, in the second qualitative stage,

a number of them were chosen to be interviewed for the purpose of fleshing out the data obtained from the questionnaire.

The study population was made up of the 76 teachers who taught the Islamic religion module in the pre- and primary school stages in Spain during the academic year 2018/2019 (UCIDE 2019). The CIE provided a list of schools at which the Islamic religion module was taught in that academic year. In Stage 1, a link to the questionnaire was sent to the teachers via Google Forms, obtaining a sample of 59 participants accounting for 78 per cent of the total number of teachers.

With the aim of protecting their anonymity, the participants requested the research team not to include the city in which they taught, since in some places there was only one teacher. The salient features of the respondents were as follows:

- 59.3 per cent were men and 40.7 per cent were women.
- 83.1 per cent were older than 36.
- 13.36 per cent only taught in the pre-school stage, while 86.4 per cent taught in both the pre- and primary school stages.
- 64.4 per cent were born in Morocco, 32.2 per cent in Spain and 3.4 per cent in Algeria.
- 55.9 per cent were second-generation immigrants, 30.5 per cent first-generation immigrants and 13.6 per cent converts to Islam.
- All had been residing in Spain for over 15 years.
- The majority of them (66.1 per cent) had been teaching Islamic religion for more than 11 years.
- As to the branch of Islam that they professed, most of them (96.6 per cent) were Sunnis and 3.4 per cent, Shias.
- 91.5 per cent claimed that they taught Sunni Islam, and 8.5 per cent, a general overview, without focusing on the Sunni or Shia branches.

In Stage 2, conducted in 2019, a total of 13 teachers were interviewed, this being the point at which theoretical saturation was reached (Valles 2014). Those teachers volunteering to be interviewed were selected according to a series of inclusion criteria (Robinson 2014), so as to form the most heterogeneous group possible: an equal number of men and women, both converts and immigrants and a different amount of teaching experience. Specifically, 46.2 per cent were men and 53.8 per cent were women, 38.5 per cent were converts and 61.5 per cent were immigrants (from Morocco and Algeria).

*3.2. Instruments*

The questionnaire was designed on the basis of a comprehensive bibliographic review of the topic (UNESCO 2017; Dzhekova et al. 2017; Christmann 2012; RAN 2016; Ranstorp 2010; Roy 2015). It had a total of five dimensions including multiple choice, open and 4-point Likert scale (1 = Not at all and 4 = A lot; 1 = Highly inadequate and 4 = Highly adequate) items.

The questionnaire's reliability was calculated using Cronbach's alpha as a measure of internal consistency, obtaining values close to the unit for both the questionnaire as a whole ($\alpha$ = 0.898) and its five dimensions.

The following dimensions were analysed (see Table 1):

**Table 1.** Questionnaire dimensions analysed.

| Dimension | Objective | No. of Items | Cronbach's Alpha |
|---|---|---|---|
| The teachers' religious beliefs | To become acquainted with the teachers' views on specific beliefs pertaining to Islam | 1 multiple choice 4 Likert scale | $\alpha = 0.844$ |
| Preventive socio-educational measures | To become acquainted with the teachers' views on the measures proposed by international bodies for preventing violent radicalism, as well as the socio-educational measures that the they themselves proposed | 2 open 9 Likert scale | $\alpha = 0.945$ |

Source: Own elaboration.

Construct validity was verified using two procedures: A. Exploratory factorial analysis (EFA). Employing Bartlett's test of sphericity and the Kaiser–Meyer–Olkin (KMO) measure of sampling adequacy, satisfactory results confirming the pertinence of performing the EFA were obtained (see Table 2). This was performed with the principal component method, obtaining saturations of higher than 0.40 for each factor (Floyd and Widaman 1995); B. Non-metric multidimensional scaling (NMDS) with the PROXSCAL program. With this method, it was possible to confirm the goodness-of-fit of the model, obtaining stress values of close to 0 and DAF and Tucker values of close to 1, all of which indicated an acceptable goodness-of-fit (Biencinto et al. 2013).

As to the study's qualitative instrument, a semi-structured interview was designed after analysing the results of the questionnaire. The initial script included 12 questions. However, the flexibility of this type of method allowed for incorporating questions that were believed to be interesting during the interviews.

**Table 2.** Construct validity.

| Scale | KMO | Bartlett's Test | | | Saturation Coefficients | % Variance | Stress and Goodness-of-Fit Measures | | | | | |
|---|---|---|---|---|---|---|---|---|---|---|---|---|
| | | $\chi^2$ | Gl | Sig. | | | NORMALISED Raw Stress | Stress-I | Stress-II | S-Stress | DAF | Tucker |
| The teachers' religious beliefs | 0.732 | 161.338 | 10 | 0.000 | 0.633; 0.858; 0.884; 0.898; 0.658 | 63.128 | 0.00012 | 0.01101 | 0.02507 | 0.00022 | 0.99988 | 0.99994 |
| Preventive socio-educational measures | 0.891 | 557.831 | 36 | 0.000 | 0.918; 0.883; 0.764; 0.731; 0.871; 0.910; 0.900; 0.869; 0.675 | 70.534 | 0.00339 | 0.05824 | 0.13110 | 0.00820 | 0.99661 | 0.99830 |

Source: Own elaboration.

*3.3. Data Analysis*

For the quantitative analysis, descriptive (percentages), correlational (contingency coefficient for bivariate analysis) and inferential statistics were employed with the SPSS v. 24 software package. For the statistical inference analysis, the non-parametric Kruskal–Wallis H test was run, after the non-normality of the sample in the study variables ($p = 0.000$) had been verified with the Kolmogorov–Smirnov (K–S) goodness-of-fit test.

For the correlational and inferential statistical analyses, the 'generation' variable was used. This included converts, first-generation immigrants (i.e., born abroad) and second-generation immigrants (with at least one parent born in Spain).

The interviews were recorded with the consent of the interviewees and then transcribed for their analysis using the Atlas. ti v. 7.5 program. To identify the analytical categories, a mixed coding process was employed in two consecutive stages following a thematic criterion (Creswell 2013). In the first stage, a deductive analysis was performed with the categories previously established on the basis of the research objectives and, in the second, an inductive analysis was performed on the topics emerging from the transcriptions.

## 4. Results

*4.1. The Islamic Religion Teachers' Views on the Interpretation of the Sacred Texts and the Concept of Jihad*

Stage 1:

Regarding the interpretation of the sacred texts (see Item 1 in Table 3), 64.4 per cent of the respondents were of the opinion that this should be literal, while 69.5 per cent held that they should be interpreted by adapting them to present times (see Item 2 in Table 3). In both cases, statistically significant differences ($p = 0.000$) were detected between the converts and the immigrants.

**Table 3.** Results of the Kruskal–Wallis H test.

|  | **Average Range** | | | | |
|---|---|---|---|---|---|
|  | **Convert** | **Immigrant** | **Second Generation** | **Kruskal–Wallis H Test** | **Significance ($p < 0.005$)** |
| Item 1 | **14.88** | **46.17** | 24.85 | 27.367 | 0.000 |
| Item 2 | **55.00** | **14.11** | 32.61 | 36.423 | 0.000 |

Source: Own elaboration.

The converts agreed more on the need to teach pupils to interpret the sacred texts by adapting them to present times and less on a literal interpretation of the Qur'an than the immigrants.

With respect to the concept of lesser jihad, most of the respondents (72.9 per cent) disagreed that it had only made sense at the time when the Qur'an was revealed.

In order to discover their views on the acts that they considered to be lesser jihad, the following options were established: (1) the Afghan resistance against the Soviet invasion; (2) the Palestinian resistance; (3) the terrorist attack against Charlie Hebdo in Paris; (4) the Iraqi resistance against the US invasion; (5) none; and (6) all.

None of the respondents considered that the terrorist attack against Charlie Hebdo was lesser jihad. The majority (61 per cent) believed that the Palestinian resistance was indeed lesser jihad, followed by 23.7 per cent who selected 'All resistance against an aggressor' (i.e., Palestinian, Afghan and Iraqi).

As to whether or not the Qur'an endorsed self-defence against unjustified aggressions, 81.3 per cent of the respondents agreed that this was indeed so.

Stage 2:

The interviews yielded results that supported those obtained in Stage 1 of the study. It was, above all, the converts who were against a literal interpretation of the Qur'an: "The Qur'an is a very

difficult text and has multiple interpretations. There're many things that cannot be taken at face value" (I[1] 10); "There're Muslims who out of fear of introducing personal aspects, let's say of distancing themselves from the classical norm, interpret it very literally" (I 6). Noteworthy is the following argument deployed by a female convert who related literal interpretations to extremism:

To my mind, an extremist Muslim is whoever takes the sacred texts at face value, without bearing in mind what the Prophet said. For instance, many say we cannot mix with Christians and, reading the history of the Prophet, you discover it isn't so. It does appear in some verses of the Qur'an, but it's referring to the fact that we shouldn't follow them. There're times that I myself don't understand the Qur'an, so I go to the imam who's the person who explains it to you (I 5).

Those who claimed that pupils should not be taught to interpret the sacred texts by adapting them to the present times, pointed out that the Qur'an conformed to all ages.

As to the concept of lesser jihad, one of the female interviewees justified why for her it had only made sense in the times of the Prophet: "At that time, Muslims didn't have to defend themselves against polytheists, but nowadays I think those elements of self-protection should be dispensed with because modern states have legal measures" (I 10).

A minority of the teachers confused the meaning of lesser jihad with that of greater jihad, the latter being the one that involves making an effort to be a good Muslim. The rest defined it as a struggle with two main purposes: (1) both self-defence against an attack and the defence of the rights and the faith of Muslims; and (2) the quest for justice. They highlighted the need to consider a series of conditions, reflected in the Qur'an, for legitimate armed struggle, while placing the accent on breaking with the definition of 'holy war' transmitted by many media outlets, which they believed was incorrect.

Those respondents who considered that none of the types of resistance included in the questionnaire could be regarded as lesser jihad held that they should be understood as political rather than religious wars: "None of this has anything to do with Islam, but with political and territorial interests" (R[2] 9). Similarly, they stressed that any act of defence that violated the rules set out in the Qur'an was not self-defence but terrorism: "They have a right to defend themselves when a power invades their territory, but not to put on a bomb vest and go to a market where there're innocent people" (R 4).

Those who considered that all resistance against an aggressor was legitimate, criticised the invasion of Arab countries by the Western powers:

No power has the right to 'impose' its values and convictions, destroying the lives of millions of people. We're talking about three societies (Afghanistan, Palestine and Iraq) destroyed in body and soul and whose recuperation we sadly won't see. Another aspect is how they carry out that resistance (R 30).

Those who believed that the Palestinian resistance could be regarded as lesser jihad referred to the ongoing oppression of the Palestinian people: "They're depriving them of their territory and assets. They're murdering children, women . . . and in light of that the only legitimate resistance that's left to those people is to defend themselves" (R 5).

As regards to what should be understood by an unjustified aggression in lesser jihad, the respondents observed that it was a very complex and far-reaching issue, which could encompass different meanings: (1) an invasion (they highlighted the examples of Afghanistan, Iraq and Palestine); (2) an act of violence; (3) oppression against Muslims and the violation and deprivation of their rights; and (4) discrimination on ethnic or religious grounds. Although, they emphasised the importance of teaching pupils how to react to these aggressions: never with a thirst for vengeance or violence: "Self-defence is a personal decision, you can lodge a complaint. It doesn't mean you should react violently" (R 6).

---

[1]  I = interview.
[2]  R = reply to an open item in the questionnaire.

*4.2. The Islamic Religion Teachers' Views on Terrorism in the Name of Islam*

Stage 2:

According to the interviewees, there was a large variety of reasons why Islam was used for violent purposes.

All the interviewees held that this was due to a deficient knowledge of Islam, which made it possible to manipulate the young with messages that they received via the Internet or other media outlets: "If we look at the information on arrests, it can be clearly seen that they're young people with a very poor knowledge of Islam" (I 4).

They chiefly highlighted the mistaken concept of jihad that they learnt from the media: "Children access the Internet and read that what they did, for example, in Charlie Hebdo is jihad. For which reason it's important they're taught the real meaning of this and other religious concepts" (I 11).

All the interviewees attached great importance to the frustration that these young people might feel and which might lead them to resort to violence. Discrimination against Muslims could also contribute to this. They also underscored identity crises, particularly among second-generation immigrants. However, they also stressed that there was no clear-cut profile.

Lastly, some thought that it was due to the West's political and economic interests in Muslim-majority countries: "The Islamic world's like a cake, and it's about who can get the biggest slice of it" (I 5); "Why are there wars where there's oil?" (I 2); "Groups like Daesh are laboratory-made. Young people who sacrifice themselves can be simple and bitter, but I'm convinced that what's behind that is high politics with geopolitical interests. Divide and rule" (I 7).

*4.3. The Islamic Religion Teachers' Views on Preventive Socio-Educational Measures*

Stage 1:

For 83 per cent of the respondents, the measures implemented in the United Kingdom, obliging teachers to detect possible 'signs of radicalisation' in their pupils to prevent violent radicalism, were inappropriate.

The most highly valued measure was to train Islamic religion teachers in order that they should be able to offer their pupils arguments in favour of learning an Islam based on peace (average = 3.63),[3] while the least valued was to demand that imams receive a unified religious education (average = 2.59).

Statistically significant differences (Kruskal–Wallis H test) in two of these items were detected depending on the generation to which the respondents belonged. Teaching the verses of the Qur'an, while taking into account the context in which they were revealed, was more acceptable to the converts (with a value H = 20.212 and $p = 0.000$; average range = 51.00) than to the immigrants (average range = 19.92). Conversely, demanding that imams receive a unified religious education was more acceptable to the immigrants (with a value H = 13.686 and $p = 0.001$; average range = 39.08) than to the teachers born in Spain (average range = 23.32).

The indicator that they believed was most adequate were identity crises (average = 3.29) with the second-generation immigrants valuing it more positively (with a value H = 50.930 and $p = 0.000$; average range = 42.50) than their first-generation colleagues (average range = 13.17).

The least valued indicator was a sudden, excessive and exclusive interest in a specific religion (average = 2). It was found (with a value H = 8.432 and $p = 0.015$) that this indicator was more inadequate for the converts (average range = 16.00) than for the immigrants (average range = 35.97).

Stage 2:

Most of the interviewees and respondents considered that the measures implemented in the United Kingdom were inadequate, since they directly stigmatised Muslim pupils: "It's subjective. It depends on what teachers understand by and know about Islam. For example, for some, a girl who

---

[3]　Averages are shown on the following 4-point Likert scale: 1 = 'Strongly disagree' and 4 = 'Strongly agree'.

goes from not wearing a hijab to wearing one is a radical" (I 5); "Many Muslim pupils will lose out. We've gone back to pigeonholing and labelling pupils" (R 11).

Others noted that if teachers were properly trained, they would see it as an adequate measure: "It's a good measure if teachers have the necessary training. On the contrary, I think that asking teachers to make that decision, without the necessary training, is totally abhorrent" (I 2).

There were arguments that justified the fact that the converts thought that the indicator "a sudden, excessive and exclusive interest in a specific religion" was the most inadequate: "When I became interested in Islam and decided to wear a hijab, my family thought I was becoming radicalised" (I 7). They stressed that what most concerned them was the sudden rejection of everything that clashed with the views of pupils (R 48), as well as considering all those who did not practice true Islam (R 17; 59; 47; 34) as "deviants" and rejecting other religions (R 14; 15; 34; 33; 7).

Some considered that a breakdown in relations was also a very subjective indicator, inasmuch as "it can indicate something else, for instance, that the person in question is depressed" (R 29).

They also noted that being excessively concerned about merely superficial issues, such as clothing, beards, etc., was another indicator of radicalisation (R 12; 30; 32; 14; 15).

The argument deployed by one of the male interviewees, who was totally against the so-called "indicators or signs of radicalisation", offers food for thought: "I think that when someone's becoming radicalised, he's more likely to conceal it. That is, he doesn't mention it or appear to be so and even possibly stops attending the mosque" (I 5).

By order of frequency, the educational measure that was considered to be the most adequate for preventing radicalisation was the teaching of Islam (f = 40)[4]. For the teachers, it was very important to provide pupils with a sound knowledge of the fundamentals of Islam, since they believed that this would make it very difficult for anyone to manipulate them: "A good knowledge of Islam serves as a filter for processing the biased information that the young obtain on the Internet" (R 16).

In this regard, they attached importance to three basic issues:

1. Teaching pupils the verses of the Qur'an adapted to the context in which they were revealed: "To clarify the meaning of the verses of the Qur'an, employed to justify terrorism and the loathing of otherness, adapted to their context" (I 25).
2. Discussing the issue of terrorism in the name of Islam in the classroom.
3. Paying attention to the teachings and messages transmitted by imams at mosques. Some of the teachers claimed that they sought the advice of their imam when in doubt. Specifically, it was the converts among their number who thought that it was important to know how to convey messages contextualised to the country in which they resided: "We need moderate imams who serve as references. Many of them don't speak Spanish and, for that reason, the young search for information on the Internet" (R 32); "There're highly qualified Spanish Muslims who could be imams and empathise with the problems that young Spanish Muslims may have" (I 6).

The second most cited educational measure was an education in both universal values (f = 20), and those pertaining to each religion forming part of the religious landscape in Spain. Lastly, they mentioned the need to educate and inform families (f = 12): "For me it's important and necessary to educate families so as to make them more aware of the information on Islam that their children may read on the Internet" (R 50).

As to social measures, the most cited was to foster the social integration of Muslims (f = 47): "A commitment to a greater social integration of Muslims to avoid exclusion. I believe a young person becomes radicalised because there's a serious problem of exclusion" (R 47). They placed special emphasis on the need to implement measures to avoid the creation of ghettos and to pay attention to

---

4　　The frequency of the categories obtained in the qualitative analysis of the interviews and the questionnaire's open-response items is shown in brackets.

possible identity problems, thus fostering a sense of belonging to Spanish society: "It's necessary that children born here feel Spanish" (I 3).

The second social measure that they proposed was the prevention of Islamophobia (f = 25), placing the accent on the negative messages and stereotypes regarding Islam and Muslims transmitted by many media outlets.

## 5. Discussion and Conclusions

In relation to the reasons behind terrorism in the name of Islam, we detected a series of factors that, according to Islamic religion teachers in Spain, might encourage the young to feel attracted to this ideology.

1. A deficient knowledge of Islam is one of the main reasons why the young are a collective susceptible to being convinced to commit acts of violence; whereby the teaching of Islam was the preventive educational measure most highly valued by the teachers. They believed that it was very important to provide Muslim pupils with a solid knowledge of the fundamentals of Islam, for they considered that this would make it more difficult to persuade them to commit acts of violence.
2. Identity crises were the second most frequently mentioned factor and also an indicator of radicalisation on which there was the greatest consensus among the teachers, especially the second-generation immigrants among their number. This could be related to the fact that it is the members of this generation who have suffered most from these identity crises, feeling that they do not belong either to their parents' ethnic culture or to the society into which they have been born (Verkuyten 2018). We concur with Hoque (2018) that this subject should be broached in the classroom, since it would be beneficial for teachers to address the diversity of (cultural, ethnic, linguistic, religious, etc.) identities with their pupils.
3. The teachers considered that it was important to pay attention to the feelings of discrimination that young Muslims may harbour, which coincides with studies that have evinced how such feelings are related to a greater vulnerability to violent radicalism (Victoroff et al. 2012). Indeed, although not everyone who suffers from discrimination acts or reacts in a violent manner, young people's frustration with a society that they believe has not met their expectations and treats them unfairly is an important factor in violent radicalisation processes (Roy 2015).
4. The teachers also referred to the impact of politico-economic factors, especially those relating to Western foreign policy with Arab-Muslim countries. For authors like McCauley (2018), these policies can lead to feelings of injustice and frustration that should be avoided. The views of the teachers participating in our study are in line with the arguments deployed by Zhirkov et al. (2014), who suggest that better relations between the West and the Arab-Muslim world would undermine social support for terrorism.

To analyse and verify the causal nature of these factors, it would be necessary and desirable to perform other specific studies. In this study, we have focused on the field of formal education. It would be advisable to perform further studies that enquire more deeply into the social measures necessary for enhancing the Muslim community's sense of belonging to Spanish society.

The fact that the teachers from immigrant backgrounds were more in favour of interpreting the Qur'an literally than the converts might have to do with the stronger links and attachment that many immigrants tend to have to their culture of origin and traditions as an important source of identification (Llorent-Bedmar and Llorent-Vaquero 2014).

Jihad is an Islamic concept that is habitually employed in a biased manner in the West and, according to Venkatraman (2007), out of context by groups like Daesh to legitimise their acts of violence. The majority of the teachers participating in our study disagreed that the concept of lesser jihad had only made sense in the socio-historical context existing when the Qur'an was revealed.

All our interviewees concurred with Rane (2019) when understanding jihad in a defensive sense of the word and, therefore, did not envisage the possibility of an offensive jihad. For most of them, the Palestinian resistance was lesser jihad, while none of them considered the attack against Charlie Hebdo in Paris or other similar acts as such, showing themselves to be diametrically opposed to the fairly widespread interpretation of the concept in the West, which confuses it with 'holy war'.

Most of the teachers considered that the Qur'an allowed self-defence in the face of injustices committed against Muslims. Taking into account that some terrorist groups present jihad as a legitimate act of defence against the suffering of the Muslim community (Tezcür and Besaw 2020), it is vital to teach pupils the socio-historical context in which the verses of the Qur'an pertaining to lesser jihad were revealed, as well as the strict conditions that should be met in order to wage jihad in self-defence (De León Azcárate 2018).

With respect to preventive educational measures, most of the teachers considered that the UK Prevent strategy was inadequate, stating that it could be counterproductive as it stigmatised Muslim pupils, in consonance with studies performed on this strategy (Kyriacou et al. 2017). Be that as it may, some indeed believed that it was adequate, provided that teachers were given the right training.

We concur with Onursal and Kirkpatrick (2019) that obliging teachers to identify pupils who may be vulnerable to violent radicalism is a controversial issue, since research on this phenomenon is not conclusive (Lynch et al. 2015; Verkuyten 2018). Furthermore, the teachers participating in our study were of the mind that the so-called "indicators of radicalisation" were very subjective, which is in line with the difficulties that teachers have when judging these signs (Moffat and Gerard 2020). In this connection, the arguments deployed by the converts in our study are especially interesting, insofar as they tended to adopt behaviours that were not considered to be normal in their environment, whereby, taking into consideration these indicators, they could be understood as being at risk of radicalisation. In particular, they were against the indicator of "a sudden, excessive and exclusive interest in a specific religion", since this was how they themselves had behaved in the initial stages of their conversion.

The preventive educational measure most highly valued by the teachers participating in our study was improving teacher training, especially as in regards to the arguments that they could deploy so as to persuade their pupils to learn an Islam based on peace. One of the least valued measures was to demand that imams in Spain receive a unified religious training. Nonetheless, it is also true that the teachers from immigrant backgrounds valued this measure more highly, something that ties in with the fact that different collectives of Moroccan immigrants in Spain are in favour of training imams at state-controlled institutions, as is the case in some Arab countries (Moreno 2017). Bearing in mind the influence that imams can have on the opinions of the young and, therefore, the problem posed by the fact that they are underqualified (Cherribi 2010), it is vital to regulate their training, while paying special attention to the social and cultural context of Spanish society.

All these educational measures should be accompanied by social measures that enhance the Muslim community's sense of belonging to Spanish society. Accordingly, it is essential to raise the awareness of the host society in order to avoid Islamophobia and, consequently, possible feelings of discrimination among Muslims.

The concern for detecting early signs of violent radicalisation, without previously considering the phenomenon's complexity and subjectivity, is giving rise to the adoption of preventive strategies, like Prevent in the United Kingdom, which we do not endorse. As neither did the teachers participating in our study agree with its implementation, we should highlight the negative consequences that some measures like this might have.

**Author Contributions:** Conceptualisation, M.N.-G., V.L.-B. and V.C.C.-D.P.; methodology, M.N.-G.; software, M.N.-G.; validation, M.N.-G.; formal analysis, M.N.-G.; investigation, M.N.-G., V.L.-B. and V.C.C.-D.P.; resources, V.L.-B.; data curation, M.N.-G.; writing—original draft preparation, M.N.-G., V.L.-B. and V.C.C.-D.P.; writing—review and editing, M.N.-G., V.L.-B. and V.C.C.-D.P.; supervision, V.L.-B.; project administration, V.C.C.-D.P.; funding acquisition, V.L.-B. All authors have read and agreed to the published version of the manuscript.

**Funding:** This paper has been written in the framework of the Research Plan of the University of Seville (VPPI-US-2016): Pre-doctoral or trainee researcher grants; and as part of the project 'Islam and peace through Muslim voices. Preventive socio-educational measures', funded by the Office of Development and Cooperation of the University of Seville

**Conflicts of Interest:** The authors declare no conflict of interest.

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
