# Peer review of "The Views on Terrorism in the Name of Islam Held by Islamic Religion Teachers in Spain"

_religions, doi:10.3390/rel11110624_

Round 1

Reviewer 1 Report

Dear authors, 

Thank you for providing me with the opportunity to read your study on teacher perceptions of Islam. I've attached a word document where I've provided feedback via track changes. Please see that attached. 

I have some overarching concerns, which I discuss below. 

  1. You did not clearly articulate a conceptual framework. 
  2. You have too many research questions, with many reading as interview questions. Revise the objectives into a purpose statement and then revise your research questions to no more than 3. 
  3. I am not convinced that your figures aid in understanding. Consider removing those and articulating the findings through themes and sub-themes. The connections are not clear between boxes, leading to confusion and misalignment.
  4. Make your headings in your findings section more descriptive. Let the reader know what you will say through your heading. 
  5. Bring forward the issue of the converts/reverts. This was a significant finding, so bring out their perspectives more. How many converts were interviewed? 
  6. A summary is needed that reminds the reader of purpose of the study and then concludes with recommendations. 
  7. I have a conceptual issue that I want to address with you. You have the assumption that more education of Muslims will thwart radicalisation. Yet there is research that shows terrorists tend to have high levels of education. You need to look into this assumption. 
  8. I also found it problematic that you place the onus of radicalisation on the students and their learning. What can the wider Spanish society do/change/modify to improve feelings of belongingness of their Muslim neighbours? 
  9. Make sure that your discussion clearly states that these are perspectives of the teachers. 
  10. You seem to be covering too much. Revise for alignment of key ideas. 

Best wishes to you as you complete the publication process. 

Author Response

  1. You did not clearly articulate a conceptual framework. 

REVIEWER 1

  1. You did not clearly articulate a conceptual framework. 

We have reorganized the sections, with an introduction and a theoretical framework subdivided into different sections to make it seem clearer.

  1. You have too many research questions, with many reading as interview questions. Revise the objectives into a purpose statement and then revise your research questions to no more than 3. 

We have changed the objectives for the purpose of the study and included three research questions: ‘It order to remedy this gap, the…’ (p. 2).

 am not convinced that your figures aid in understanding. Consider removing those and articulating the findings through themes and sub-themes. The connections are not clear between boxes, leading to confusion and misalignment.

We have eliminated the figures, including the information that we consider relevant in the text.

  1. Make your headings in your findings section more descriptive. Let the reader know what you will say through your heading. 

We have modified the subtitles of the findings section in a more concrete and descriptive way.

  1. Bring forward the issue of the converts/reverts. This was a significant finding, so bring out their perspectives more. How many converts were interviewed? 

We have specified the number of Muslim converts who were interviewed: ‘Specifically, 46.2 per cent were…’ (p. 5).

We have included some quotes in the results section: ‘it was the converts among their number who…’ (p. 9).

  1. A summary is needed that reminds the reader of purpose of the study and then concludes with recommendations. 

We have included the purpose of the study and recommendations.

  1. I have a conceptual issue that I want to address with you. You have the assumption that more education of Muslims will thwart radicalisation. Yet there is research that shows terrorists tend to have high levels of education. You need to look into this assumption. 

We have included a paragraph in the "background" section on this issue, in which we discuss different positions: ‘The relationship between educational attainment and terrorism is not that clear. Authors…’ (p. 3).

  1. I also found it problematic that you place the onus of radicalisation on the students and their learning. What can the wider Spanish society do/change/modify to improve feelings of belongingness of their Muslim neighbours? 

We have made a clarification on page 11: In this study, we have focused on…’. We have also included a paragraph on page 12: ‘All these educational measures should be accompanied by social measures that …’

  1. Make sure that your discussion clearly states that these are perspectives of the teachers. 

We have reviewed the entire section with this in mind.

  1. You seem to be covering too much. Revise for alignment of key ideas. 

We have eliminated some results that we felt were not relevant enough and did not exactly answer the research questions.

We have also eliminated some issues that we considered less relevant in the discussion.

Reviewer 2 Report

Overall, the research follows a familiar pattern, but breaks new ground in its focus. It could be strengthened in a number of ways.

First, to what extent does the research reflect the size and variety of the ethno-Muslim communities in Spain? This data can usually be gleaned from the Brill Islam in Europe Yearbooks.  Does it matter that converts are probably over-represented?

Secondly, there is clearly a problem with non-contextualized Islamic teaching by imams lines 429-431: more could be said about this, which, in part, explains the frustration which leads second generation Muslims to 'sheikh google'! In this regard, the author[s] might well read an excellent analysis of the ethos and sermons of fifteen Moroccan mosques in Amsterdam (albeit of an earlier period) in Sam Cherribi, "In the House of War, Dutch Islam Observed" (OUP, 2010). Cherribi is himself a Moroccan and was an MP in  Holland, the book was based  on his PhD.

Thirdly, this study deals with two generations of Spanish Muslims.  The authors might want look  at Aminul Hoque' "British Islamic Identity: Third Generation Bangladeshis from East London" (2015) - this is hopeful study; it might be used, for example to  suggest a possible convergence between their findings about converts and  the next generation of Spanish Muslims.

There are several minor changes in English which would clarify meaning. line 91: substitute 'indiscriminately' for 'indistinctly'; 139 something cannot be 'obligatory' & 'elective' at the same time; 152 'degree' for ''decree'; 161 sentence unclear; 409 'deviants'  for 'deviates'.

Lines 223-32 disrupts flow of argument -  as  well as very technical - so might  be better in a  footnote.

Author Response

REVIEWER 2

First, to what extent does the research reflect the size and variety of the ethno-Muslim communities in Spain? This data can usually be gleaned from the Brill Islam in Europe Yearbooks.  Does it matter that converts are probably over-represented?

Our study focuses exclusively on the Islamic religion faculty in Spain, not on the Muslim community in Spain. In this case, we have addressed the population as a whole, obtaining a high response rate, so we estimate that it is representative of that population.

Secondly, there is clearly a problem with non-contextualized Islamic teaching by imams lines 429-431: more could be said about this, which, in part, explains the frustration which leads second generation Muslims to 'sheikh google'! In this regard, the author[s] might well read an excellent analysis of the ethos and sermons of fifteen Moroccan mosques in Amsterdam (albeit of an earlier period) in Sam Cherribi, "In the House of War, Dutch Islam Observed" (OUP, 2010). Cherribi is himself a Moroccan and was an MP in  Holland, the book was based  on his PhD.

We have introduced a quote from Cherribi in the discussion section: ‘Bearing in mind the influence that imams can have on the opinions of…’ (p. 12).

We have also gone a little deeper into this topic on page 10: ‘Some of the teachers claimed that they sought the advice of their…

Thirdly, this study deals with two generations of Spanish Muslims.  The authors might want look  at Aminul Hoque' "British Islamic Identity: Third Generation Bangladeshis from East London" (2015) - this is hopeful study; it might be used, for example to  suggest a possible convergence between their findings about converts and  the next generation of Spanish Muslims.

We have used the quote on page 11: ‘We concur with Hoque (2018) that…

There are several minor changes in English which would clarify meaning. line 91: substitute 'indiscriminately' for 'indistinctly'; 139 something cannot be 'obligatory' & 'elective' at the same time; 152 'degree' for ''decree'; 161 sentence unclear; 409 'deviants'  for 'deviates'.

We have made the requested changes marked in red. The paragraph in line 161 has been deleted as it did not provide relevant information for the study.

Lines 223-32 disrupts flow of argument -  as  well as very technical - so might  be better in a  footnote.

We have included these arguments in a footnote on page 6.

The format of the first row in Table 3, "Average range", is a bit dubious as to which part of the rest of the table it covers, please revise

We have modified the format of table 3 on page 7.